# Recent Advances in Counterfeit Art, Document, Photo, Hologram, and Currency Detection Using Hyperspectral Imaging

**DOI:** 10.3390/s22197308

**Published:** 2022-09-26

**Authors:** Shuan-Yu Huang, Arvind Mukundan, Yu-Ming Tsao, Youngjo Kim, Fen-Chi Lin, Hsiang-Chen Wang

**Affiliations:** 1Department of Optometry, Central Taiwan University of Science and Technology, No. 666, Buzih Road, Beitun District, Taichung City 406053, Taiwan; 2Department of Mechanical Engineering, Advanced Institute of Manufacturing with High Tech Innovations (AIM-HI), Center for Innovative Research on Aging Society (CIRAS), National Chung Cheng University, 168, University Rd., Min Hsiung, Chia Yi 62102, Taiwan; 3Department of Mechanical Engineering, Far Eastern University, P. Paredes St., Sampaloc, Manila 1015, Philippines; 4Department of Ophthalmology, Kaohsiung Armed Forces General Hospital, 2, Zhongzheng 1st Rd., Lingya District, Kaohsiung City 80284, Taiwan

**Keywords:** hyperspectral imaging, forgery detection, artwork authentication, document authentication, counterfeit currency detection, photo authentication, hologram authentication

## Abstract

Forgery and tampering continue to provide unnecessary economic burdens. Although new anti-forgery and counterfeiting technologies arise, they inadvertently lead to the sophistication of forgery techniques over time, to a point where detection is no longer viable without technological aid. Among the various optical techniques, one of the recently used techniques to detect counterfeit products is HSI, which captures a range of electromagnetic data. To aid in the further exploration and eventual application of the technique, this study categorizes and summarizes existing related studies on hyperspectral imaging and creates a mini meta-analysis of this stream of literature. The literature review has been classified based on the product HSI has used in counterfeit documents, photos, holograms, artwork, and currency detection.

## 1. Introduction

Counterfeiting and forgery are salient issues which plague the global economy. It is estimated that losses from counterfeit checks in the U.S. alone have exceeded $20 billion, whereas public losses from counterfeit dollar notes were estimated at $80 million [1]. Moreover, the cost of re-designing the currency every 7–10 years as the counter-forgery measurement must also be considered [2]. While not as common as currency, artworks are also vulnerable to forgery, as they are usually authenticated by experts, thereby leading to much human error [3,4,5]. Considering the enormous size of the global art market amounting to €51 billion, a more systematic and scientific method of artwork authentication is therefore necessary [6]. In contrast, document forgeries are considered major issues in police departments because they cause economic problems and pose an actual danger to the public as they can be involved in crime [7,8].

Methods used in the forgery technologically evolve on a constant basis and are becoming harder to verify the legitimacy of a document [9]. The human eye can only detect colors that are combinations of red, blue, and green (RGB) [10]. Meanwhile, hyperspectral imaging (HSI) detects lights with any wavelength ranging from ultraviolet (UV) to far-infrared (FIR) ones. Additionally, it can also distinguish between the two different materials emitting identical RGB values, based on their spectral fingerprints [11].

HSI has been successfully applied in various applications. In medical fields, its application varies from visual aids for diagnostic purposes, and clinical analysis, to cancer detection [12,13,14,15,16,17,18,19,20,21,22,23,24,25,26,27,28,29,30,31]. In agriculture, it is often used for quality control and inspection purposes [32,33,34,35,36,37,38,39,40,41,42,43]. In military applications, it is often used for countermeasure detection and recognition of camouflaged targets [44,45,46,47,48,49,50,51,52,53,54,55,56,57,58]. Material sciences uses HSI to detect and identify the materials using their spectral signature [59,60,61,62,63,64,65,66,67,68,69,70,71]. In remote sensing, it is used for geological exploration and soil characterization from a distance [72,73,74,75,76,77,78,79]. HSI paired with its remote sensing capability is often used in the field of astronomy for astronomic observation and space surveillance purposes [80,81,82,83,84,85,86,87,88,89,90]. In addition to these, environmental applications such as drought stress measurement, pollution detection, water resource analysis, space science, and vegetation monitoring also prove HSI’s reliability [91,92,93,94,95,96,97,98,99,100,101,102,103,104].

However, only a few studies have used HSI for artwork authentication, document forgery detection, counterfeit currency detection, image authentication, and hologram authentication. One artwork authentication technique that uses HSI is pigment identification [105]. It identifies the pigment used in the painting and inspects if the pigment used is chronologically accurate. In document forgery detection, HSI is used to differentiate the different inks and their aging [106,107,108,109]. Meanwhile, when detecting a counterfeit currency, HSI builds a spectral library of authentic banknotes, which is then compared with the hyperspectral fingerprints of banknotes [110]. For image authentication, HSI differentiates the brand of the film that the image was taken with using the hyperspectral fingerprint of each film [111]. Hence, two images taken from two different films are merged become easily detectable. As per the hologram authentication, HSI can obtain the hyperspectral fingerprints of the hologram’s reflection from different incident angles, which can then be compared with the hologram in question [112,113].

With the increase of HSI application in forgery detection, this review aims to provide a foundation for the application of HSI in forensic image analysis by deducing frequent issues and comparing data based on its numerical results. Hence, it focuses on different applications of HSI for the detection of forged artwork, documents, holograms, currency, and image detection.

## 2. Criteria for Study Selection

The studies included in this review must have had definite numerical accuracy. All research must have been written in English and have been published in SCI and Scopus indexed journals. All research must have been published in journals with impact factors of more than 3 and H-Indexes above 50. The study must have been published within the last 5 years. Narrative reviews, studies with incomplete data, systematic review/meta-analysis, comments, proceedings, or study protocols were excluded and only a selected number of conference papers with complete data and conclusions were included. See Appendix A to find the complete flowchart of study inclusion process of this review.

## 3. HSI for Artwork Authentication

Artwork authentication is a field where HSI is used as means of forgery detection technology [114]. Due to their nature as cultural heritages, artworks cannot simply be put through any form of authentication process except for those considered as non-destructive testing (NDT). Before HSI, only a handful of NDTs, such as Fourier transform infrared (FTIR), Raman spectroscopy, and X-ray fluorescence, were used for artwork authentication [115,116,117]. Therefore, the introduction of HSI would provide further variety to the possible options that can be used for artwork authentication.

Identifying pigments in the painting is one method which has been traditionally used for artwork authentication [118]. While it may not be able to prove the authenticity of a certain artwork, it can, however, prove its inauthenticity when the chronologically incorrect pigment is detected from the drawing [119]. Polak at el. used an active, laser-based, MIR hyperspectral imager (Firefly IR) and a passive hyperspectral camera operating in NIR (Red Eye 1.7) to capture the images of paintings. Spectral data of different pigments obtained from imaging the paintings were then used to construct a spectral library. Afterwards, a classification algorithm was developed and trained. For this research, Support Vector Machine (SVM) was chosen as the algorithm [120,121,122]. Once fully developed and trained, the algorithm was tested in a controlled laboratory environment and yielded a 67% (6 out of 9) and 78% (7 out 9) accuracy in pigment identification using data from Firefly IR and Red Eye 1.7, respectively.

Mirroring the spectral library constructed in the research completed by Polak et al., Casini et al., created a reference database of red lake pigment using HSI [123]. Cochineal and brazilwood paints were reconstructed through a historically accurate process and was captured by the HSI device operating in the Visual-NIR (VNIR) range. To compare results with the method that was commonly used at the time, they created a reference database of the same pigments using fiber optic reflectance spectroscopy (FORS). Subsequent comparison showed that the data acquired with HSI were accurate and greatly coincided with that of FORS. However, no exact numerical data was provided by the researchers.

Meanwhile, Daniel et al., used HIS for material identification and mapping of paintings in the Museum of Zaragoza [124]. To test the analytical suitability of different HSI systems, two systems using “pushbroom” and “mirror-scanning” were evaluated in this research, respectively, where both systems operated in otherwise identical parameters. Spectral data of both systems were captured in range of VNIR and processed using spectral angle mapper (SAM). They concluded that while HSI was applicable for artwork analysis, a new algorithm will be required to overcome some interpretation difficulties.

Likewise, Deborah et al. attempted to address crack detection in paintings with spectral processing expressed in a full-band or vector approach, as other available approaches at the time of the research were either processing the information in a marginal way or post strong data reduction [125]. Therefore, a multivariate top-hat transform, referred to as spectral convergence mathematical morphology (SCMM), was developed and compared in an experiment against two pre-existing methods, namely grayscale top-hat on a distance-map (DM) and marginal top-hat (Marg.). Results illustrated that while it was robust in the crack detection, they did not show much of an improvement in comparison to the existing approaches.

Wang et al. proposed an HSI-feature based fusion method in identifying fake modern Chinese paintings [126]. An HSI camera operating in the range of 400–900 nm was used to scan and differentiate between fake and real Chinese paintings. Then, spectral features were extracted with the use of singular spectrum analysis (SSA), while spatial features were extracted with the use of both principal component analysis (PCA) and convolution neural network (CNN). Once through, features were classified with the use of SVM wherein the proposed method yielded an accuracy of 84.6% by averaging the 10 test results of classification of random 2500 samples out of 5000 samples.

Another study that utilizes the pigment analysis approach in artwork authentication using HSI was completed by Grabowski et al. [127]. One of the most difficult challenges of pigment identification is that of similar color and elemental composition [105]. Therefore, the current study focused on the development of an algorithm for distinction among different pigments that have similar elemental compositions. The HSI camera used in the data gathering process operates in a push-broom geometry and collects 256 spectral bands in the range of short wavelength infrared (SWIR). The accuracy of the researchers’ algorithm is tested by 4 different types of paper painted with 5 different pigments in a checkered pattern. The experiment yielded an overall accuracy of 91.16%, 89.76%, 62.83%, and 79.36% for each type of paper, respectively. While most of the combinations showed over 50% of accuracy, Chrysocolla on oil paper and Egyptian green on oil canvas were found to yield less than satisfactory results.

Table 1 shows the comparison of studies classified under artwork authentication in terms of year published, dataset used, acquisition range, processing method used, and accuracy. Based on the years of publication, it can be inferred that only few advancements have been made recently. All studies classified under artwork authentication included infrared (IR) wavelength in their acquisition range. Use of IR range can help improve the visibility of concealed features that cannot be seen with naked eye as well as help distinguish pigments with similar hue [128,129]. Evidently, most of the studies employed multiple processing methods. The treatment of an HS image is completed in multiple stages and different processing methods can be employed in each stage [130,131].

## 4. HSI for Document Forgery Detection

HSI is used as a means of document forgery detection. Similar to artworks, destructive examinations are generally discouraged for documents to avoid the compromise of its originality [132]. One most commonly used document forensic analysis method is chemical analysis, such as thin-layer chromatography (TLC), which is destructive and invasive [133]. To overcome this, researchers have come up with various NDT and non-invasive methods to replace the previous method, with the most widely used being Raman spectroscopy due to its accuracy [134]. his, although not as accurate as Raman spectroscopy, is comparatively more efficient in image mapping [135]. Therefore, using HSI for document authentication possibly enhances the currently used methods.

C.S. Silva et al. explored the use of HSI in the near-infrared range (HSI-NIR) in the application of forensic analysis of document forgery. To evaluate its accuracy, they employed three different types of simulated forgeries, namely line crossing, obliterating text, and adding text. Prepared sample images were mapped at the spectral resolution of 6.3 nm and spatial resolution of 10 μm, while the selected range was at 928–2524 nm. For data-processing, PCA and multivariate curve resolution-alternating least squares (MCR-ALS) were used for obliterating and adding text, while MCR-ALS and Partial Least Squares-Discriminant Analysis (PLS-DA) were used for crossing lines. The experiment yielded 43%, 82%, and 85% accuracy for the obliterating text, adding text, and crossing lines forgeries, respectively.

In a study by J.F. Pereira et al., HSI was used in both NIR and middle infrared range (MIR) [136]. A total of 16 different pens were used as samples and PCA and Projection Pursuit (PP) were used as data processing methods. The accuracy was tested by matching the numbers written from different pens on both white paper and bank check paper with sample 2 cm straight lines drawn by each pen. Using his-MIR, PP, and PCA showed an accuracy of 97.5% (73 out of 75) and 87.5% (60 out of 75), respectively whereas using HSI-NIR only yielded an accuracy of 83.3% (5 out of 6) and 76.7% (21 out of 36), respectively. When HSI-MIR was used complementarily with HSI-NIR, both methods yielded an accuracy of 90%. They then concluded that the last configuration was best based on practicality.

Z. Khan et al. also demonstrated the use of HSI to detect mismatching inks in a handwritten note [137]. Instead of pre-selecting the band, a novel joint sparse band selection (JSBS) technique was developed to help select the most informative band for forgery detection. Likewise, an end-to-end camera-based document HSI system was specifically developed for mapping. Samples were not created but were instead acquired from a database of hand-written notes. When all bands were selected, the algorithm yielded detection accuracy of 75.4% and 74.7% for blue and black inks, respectively. However, the use of JSBS lead to an accuracy yield of 86.7% and 89%, respectively.

Another study completed by Khan et al., proposed a deep learning method for ink mismatch detection in HS document images [138]. HS images from the dataset were reshaped in CNN-friendly image formats with the help of deep learning methods and were then fed into the CNN for classification. This new method yielded detection accuracies of 98.2% and 88% for blue and black ink, respectively.

Luo et al. tried to address the major limitations in the detection of ink mismatch, which required prior knowledge on the number of inks to be distinguished and uniformity in relative proportions in the inspected image [106]. These limitations were addressed using anomaly detection combined with unsupervised clustering and was then put into test. This new method yielded a detection accuracy of 89.0% and 82.3% for blue and black ink, respectively.

A.R. Martins et al. speculated that determining the chronological order of crossed lines was a recurrent problem in forensic analysis of documents [139]. To provide easy to execute analysis protocol for this problem, the hyperspectral mode of the VSC6000 was used for mapping. The sample used included a total of 49 crossings drawn on white paper from 7 different brands of blue ballpoint pens, while the band selected was from 400 to 1000 nm. HYPER-Tools was also used for analysis, while univariate analysis (UA) and MCR-ALS were the processing used herein. The developed protocol determined the chronological order of 31 out of 49 crossings (with an accuracy of 63%).

Table 2 shows the comparison of studies classified under document authentication in terms of year published, dataset used, acquisition range, processing method used, and accuracy. It can be inferred that the application of HSI in document forgery detection remained relevant to the current research trend as some studies were recently published. Acquisition range used in this topic varied, but stayed within the most common spectral ranges from VIS to IR. Over half of the studies classified under this topic employed PCA as part of its processing method. Dimension reduction using PCA helped with speeding up the following processes and increased the overall processing performance [140].

## 5. HSI for Counterfeit Currency Detection

HSI is rarely used in the field of currency forgery detection because there are other quicker and easier methods of fake currency detection, such as UV light and digital signature [141,142]. Meanwhile, currency forgery detection with HSI, while slower and much more complex, does not require the presence of security features to detect fake currency because it utilizes spectral data taken from image of said currency instead of security features. Therefore, his may provide an alternative and possibly more efficient method in detecting currencies, which lack security features, such as coins and outdated banknotes.

S. Baek et al. classified 20 different denominations of (such as the EU Euro, Indian Rupee, US Dollar) with the help of low-resolution multispectral images [143]. Contact image sensor (CIS) was used for the image acquisition using 6 different wavelengths covering RGB to IR channels. The algorithm first sorted out obvious fakes through global classification then checked the security feature of remaining banknotes using local feature classification. Same samples were classified with the method discussed in the study completed by Kang et al., and comparisons were then made. As a result, it yielded a 99.89% (27,484 out of 27,764) classification accuracy, while the method proposed by Kang et al. only yielded 98.66% (27,392 out of 27,764).

Kang et al. proposed a counterfeit banknote detection system using multispectral images in visual (VIS) and IR spectrum [144]. Banknotes were divided into various blocks and features were extracted from each to reduce processing time. It was classified post data processing with the help of Gaussian ML classifier. The experiment yielded 99.97% (8546 out of 8549) of accuracy.

Correia et al. developed in their study a portable NIR spectroscopy device for the purpose of discriminating authentic and counterfeit Brazilian real banknotes [145]. In total, 11 different regions of interest were selected and analyzed with PCA and PLS-DA, with the latter showing 100% (12 out of 12) efficiency in distinction.

Vila et al. proposed the development of a fast and non-destructive procedure for characterizing and distinguishing between original and fake Euro notes using attenuated total reflectance (ATR)-FTIR [146]. Next, 4 different regions of interests were selected, and data was processed using GRAMS 32 Software and PCA. Upon completion, the proposed procedure turned out to be fast, non-destructive, and robust.

H.T. Lim and V.M. Murukeshan explored the possibility of applying HSI for classification and authentication of polymer banknotes by building spectral library. A pushbroom HS imager was used to scan the region of interest (ROI) of sample banknotes. PCA was then applied to the images to plot 99% confidence ellipses, which were used as classification criteria. A classification map drawn through this process closely matched the actual features of selected ROI. However, the actual detection accuracy using the spectral library acquired remains to be explored.

J.M. del Hoyo-Meléndez et al. attempted to authenticate the outdated banknotes from years 1932 and 1934 using different techniques [147]. As per HSI, a SPECIM HS system working in a pushbroom was set-up and collected 776 spectral bands with a 400–100 nm range being used. The images were then corrected for dark current and normalized using a white reference and were then processed using Envi 5.0 software. HSI analysis revealed that different types of papers were used to print the banknotes and observe the clear spectral differentiation from two suspicious banknotes. The study did not, however, test detection accuracy.

Table 3 shows the comparison of studies classified under counterfeit currency detection in terms of the year published, dataset used, acquisition range, processing method used, and accuracy. While most studies under this topic were published relatively recently, only a few were presented with definite numerical results. Acquisition ranges and processing methods seemed to follow the trend of studies classified under document forgery detection. Contrasting these studies however, processing methods employed in studies classified under counterfeit currency detection seemed to focus on simplifying and speeding up the process rather than on accuracy. This may have been due to practicality, as other methods used in counterfeit currency detection only took a fraction of a second to yield the result [148,149].

## 6. HSI for Photo Authentication

The photo forensics process was often as simple as observing the photo carefully with the naked eye since simply focusing on such geometric, optical, and physical features may lead one to a clear conclusion [150]. Even when the forgery was completed flawlessly, the pixel level inspection almost always revealed the inconsistency hidden within the image [151]. That said, the application of HSI in the field of photo forensics may seem unnecessary to some. However, the expansion of variety of the possible options may introduce furtherment of its techniques.

A. Tournié et al. used NIR HS signatures of chromogenic color photographs to identify the manufacturers of each film [111] with chromogenic color photographs from 3 different manufacturers (Agfa, Fuji, Kodak) chosen as samples. The range used was 1000–2500 nm. Data was then processed using linear discriminant analysis (LDA) and (PCA). Depending on the paper type, the research yielded an identification accuracy of 82%–96% (114 out of 118, 127 out of 148, 70 out of 85).

Leshem et al. proposed that by using the HSI, an enhanced face recognition technology could be developed to overcome the current vulnerability that was inherent to 2D image [152]. Multiple layers could be created from a single HS image and encrypting each layer and creating a binary string based on this encryption generated a unique image signature. Said signature could then be used in facial recognition processes by comparing the image-signature of registered facial image with the newly input facial image.

A. Martins et al. also proposed that NIR spectroscopy and multivariate analysis could be used to date fiber-based gelatin silver prints [153]. PCA and spectral interpretation were used to determine the correlation between the print date and film’s composition. After analyzing a total of 152 film stills from 1914 to 1986 using PLS, they successfully predicted the printing date of 66 film stills with only 6 years of prediction error.

M. Picollo et al.’s review explained how HSI was used in the Memoria Fotografica project to restore photographs [154]. It started by analyzing the photographic materials such as dyes and emulsions using HSI. For the acquisition, a range of 400–900 nm was used. Uniform manifold approximation and projection for dimension reduction (UMAP) was then used to optimize the spectroscopic study of acquired images. While this did not immediately concern image authentication, given the similarities of photographic material analyzation and artwork pigment analyzation, its potential in the field of photo forensics may also be explored.

Table 4 shows the comparison of studies classified under photo authentication in terms of year published, dataset used, acquisition range, processing method used, and accuracy. Acquisition ranges and processing methods used in these studies follow the trend of studies classified under document forgery detection and counterfeit currency detection. There were a limited number of studies available and only one of the studies was presented with definite numerical results. A lack of research interest may be due to availability of alternative technologies. Tampered images are easily detectible even without the help of HSI and alternative methods are often cheaper and faster in comparison [155,156,157].

## 7. HSI for Hologram Authentication

Rainbow holograms have been on the market for a considerable amount of time but remain widely used for security and forgery detection purposes [158]. However, over time, the technology for hologram production became readily available and the hologram itself became a subject of forgery [159]. Various techniques and anti-counterfeit measures such as fine-grained glass, image swapping, random optical phase retardation, and use of specific patterns have been introduced over the years to overcome this problem. However, each measure adds more complexity and cost to the production and verification, therefore making the use of hologram more obsolete than before [160,161,162,163]. That being said, the introduction of HSI to hologram authentication may provide a cost-efficient and effective alternative to other currently used methods.

S. Sumriddetchkajorn et al. proposed to develop an HSI-based optical structure for hologram authentication on credit cards. Hyperspectral images of credit cards were taken multiple times, and each time, the light source was placed at a different incident angle. Mapped images were then processed using a feed-forward backpropagation neural network (FFNN) with 38 genuine and 109 counterfeit credit cards being used as the data set. The developed structure was able to yield false rejection rates of 5.26% and 0.92% for genuine and counterfeit credit cards, respectively (94.74% and 99.08% accuracy, respectively).

D. Soukup et al. presented a mobile setup they designed for photometric hologram acquisition [164]. A right-light was used as the light source in the image acquisition process, whereas a mobile device was used as the data processor to process the acquired images. A new analysis algorithm was then developed to capture and compress the essential appearance properties of holograms with the use of deep learning. While an already existing photometric hologram descriptor requires both genuine and fake samples, the researchers’ design only requires genuine samples to configure its parameters. Nevertheless, it showed better reliability in fake detection.

No table was prepared for hologram authentication as only two outdated studies were classified under this topic. Lack of research interest in this topic may be due to limitations in its application as security holograms are primarily implemented as an anti-forgery technology that can be examined with the naked eye [165]. Other cheaper, easier, and faster countermeasures against hologram counterfeiting similar to photo authentication may also be the reason [162,166,167].

## 8. Discussion

Reviewing the applications of HSI in the field of forgery and tampering detection reconfirms the vast potential of HSI that is yet to be realized in this field. Use of HSI provides detailed spectral data, which, with the help of well-configured algorithms, can be used to reliably detect forgeries and tampering without the need of destructive and invasive procedures that are conventionally employed [168]. This is well-represented in the research where HSI is applied for the purpose of counterfeit currency detection. Overall, these studies yielded an average accuracy of 99.97%. This was due to the abundant anti-counterfeit measures present in banknotes which acted as region of interests for expeditious feature extraction and allowed classification algorithms to perform much accurate classification [169,170,171,172]. However, it is also noteworthy that this result does not necessarily imply the superiority of HSI over other methods in counterfeit currency detection in terms of detection accuracy because results from existing studies using other methods also show similarly high accuracy [173,174,175,176,177,178,179,180,181,182,183]. Performing a comparative analysis of different counterfeit currency detection methods may soon provide HSI’s effectivity relative to other methods. 

Another factor that greatly affected the detection accuracy of HSI was wavelength selection. Selecting broader wavelengths across the spectrum usually yielded higher detection accuracy and it was also evidenced in this review as those studies which selected multiple wavelength ranges yielded the highest average accuracy of 92.87%. That said, a random acquisition of inordinate volume of data may result in a mostly redundant dataset, which can render the processing of these data virtually ineffective. Moreover, processing a large volume of data can be both costly and time-consuming [184,185,186]. Selecting an appropriate wavelength for the given purpose, therefore, is essential in maximizing its efficiency. Based on the previous studies reviewed on this paper, most commonly selected wavelengths for the purpose of artwork and document authentication belongs to the IR region. It is due to the fact that most of the fake artwork, forged, and tampered documents are made to be nearly indistinguishable with the naked eye and, therefore, usually return spectral signatures similar to that of the authentic counterpart when in VIS region. While some studies selected broader wavelengths extended until the MIR region in an attempt to acquire a detailed spectral signature, most of the studies selected a wavelength that only extended until the NIR region due to the operating limitation of the equipment. For counterfeit currency detection on the other hand, the most commonly selected wavelengths belonged to VIS region. Counterfeit currency detection is a time-critical task that requires processing of often very large samples in a short span of time. In order to achieve this, only a narrow range of wavelength must be selected to minimize the processing time. Since anti-counterfeit measures presented on a banknote are mostly designed to be verifiable with the naked eye, selecting wavelengths that only belong to VIS may be sufficient [187]. While there are clear trends in wavelength preferences depending on their purpose, not a lot of studies have made a comparison between different wavelengths to test their effectivity, let alone specified the parameter for their wavelength selection. Further research in this prospect in the future may provide a clearer picture on the effectivity of wavelengths depending on their purpose.

An appreciable amount of studies incorporated in this review were concerning the introduction of novel processing method in this application. A lot of studies have incorporated machine learning algorithms in their image classification but the type of algorithm used were different depending on the given parameters. In studies where sample size is relatively limited, parametric algorithms such as MLC and LDA were commonly used. These algorithms yield higher classification accuracy when the sample size is limited but are incapable of handling complex dataset [188]. In studies where sample size is relatively larger however, non-parametric algorithms such as SVM were commonly used. Unlike parametric algorithms, non-parametric algorithms can classify complex dataset such as those incorporating non-HSI data. However, their performance can be affected when heterogeneity is found in data classifications [189]. For the dimension reduction algorithm, PCA, regardless of the parameters, was the most commonly used algorithm. This may be due to its effectivity in dimension reduction, which is completed by eliminating the correlation and similarities, leaving only the significance. However, it is only effective when the pre-processing of data is completed correctly. Otherwise, noises inherent in data may result in false significance [190]. In addition to these findings, it was also observed that combining multiple dimension reduction algorithms yielded higher accuracy.

Computing the average accuracies for each year the studies were published, studies published in 2018 had the highest average accuracy of 96.95%, followed by 2016 with 84.30% accuracy, 2015 with 82.85% accuracy, 2017 with 78.02% accuracy, 2014 with 69.67% accuracy, and 2019 with 63% accuracy. A sudden surge of accuracy in year 2018 and sudden dip in 2019 were observed, but upon closer inspection, it was revealed that only five accuracy values were used in the calculation of average accuracy in 2018. Moreover, only one accuracy value was considered in calculation of 2019’s average accuracy. Therefore, no particular trend concerning years in which the studies were published was found. While the latest research incorporated in this review was published in 2020, it must be noted that the vast majority of recent studies that were available were excluded following inclusion and exclusion criteria. New studies are continuously being introduced to this field, including the follow-up studies of research incorporated in this review [191,192]. A more detailed breakdown of the accuracies, including those of each topic classification, wavelength, processing method, as well as their calculations can be found in Appendix A.

Despite the fact that HSI had previously proven its efficacy in the real world setting, it was still not as routinely used for the purpose of forgery and tampering detection as other traditional methods. This may be attributed to its relative inferiority in availability, convenience, cost, turnaround time, and reliability. For example, while the chromatography system may be as costly as the HSI system, it may be more accessible as chances are many of forensic science labs are already equipped with one. It would take less time and effort as it would not require the process of making the algorithm and processing the data. It also would yield more accurate result. Similarly, UV sensors, which only cost a fraction of an HSI system, could detect counterfeit currency just as accurately, if not more so than, as a HSI system would. That said, distinctive features of his, such as non-invasiveness and spectral library, still grant a unique advantage for HSI. Therefore, with continuous research for more effective application, and constant advancement of HSI technology in general, it has potential to eventually overcome its inferiority and prove its practicality.

## 9. Conclusions

While not all topics under HSI application in forgery and tampering detection displayed a promising performance, prospected use of it remains largely optimistic. This review reveals that HSI application in counterfeit currency detection exhibited outstanding detection accuracy. Further related studies, such as comparative research between other counterfeit currency detection technologies and HSI, may be useful in evaluating the efficacy of HSI in such application. Moreover, selecting an appropriate wavelength can further enhance the performance of HSI in application of forgery and tampering detection. This is evidenced by the fact that when multiple range selection led to its detection accuracy greatly increasing. If selecting multiple ranges is unfeasible due to the limitations, selecting a specific spectral range based on its purpose also displayed high performance. It also found that using a combination of two or more machine learning algorithm greatly increased the detection accuracy, otherwise, no interdependence was found in this review between the processing method used and the performance. Although the practicality of HSI application in forgery and tampering detection is inadequate at the moment, continuous improvement will eventually prove this statement otherwise given its potential and unique advantages.

## Figures and Tables

**Table 1 sensors-22-07308-t001:** Comparison of studies (Artwork).

Authors	Year	Dataset	Range (Acquisition)	Methods (Processing)	Accuracy
Polak et al.	2017	Own dataset	MIR (Firefly IR)	PCA, SVM	67%
			NIR (Red Eye 1.7)	PCA, SVM	78%
Casini et al.	2015	Own dataset	VNIR	Customized Software	N/A
Daniel et al.	2016	CNR-IFAC open-access on-line database of reflectance spectra	VNIR	SAM	N/A
Deborah et al.	2015	Own dataset	HSI-ALL	DM	N/A
				Marg.	N/A
				SCMM	N/A
Wang et al.	2016	Own dataset	VNIR	SSA (Spectral-Only)	80.6%
				PCA (Spatial-Only)	72.5%
				Combination	84.6%
				CNN (Spatial-Only)	58.4%
Grabowski et al.	2017	Own dataset	SWIR (Tempera canvas)	Own Algorithm	91.16%
			SWIR (Tempera paper)	Own Algorithm	89.76%
			SWIR (Oil canvas)	Own Algorithm	62.83%
			SWIR (Oil paper)	Own Algorithm	79.36%

**Table 2 sensors-22-07308-t002:** Comparison of studies (Documents).

Authors	Year	Dataset	Range (Acquisition)	Methods (Processing)	Accuracy
Silva et al.	2014	Own dataset	NIR	PCA, MCR-ALS (Obliterating)	42%
				PCA, MCR-ALS (Adding)	82%
				MCR-ALS, PLS-DA (Crossing)	85%
Pereira et al.	2016	Own dataset	MIR	PP PCA	97.5% 87.5%
			NIR	PP PCA	83.3% 76.7%
			Combination		90%
Khan et al.	2015	UWA Writing Ink Hyperspectral Image Database	JSBS	JSPCA (Blue ink)	86.7%
				JSPCA (Black ink)	89%
			VIS	JSPCA (Blue ink)	75.4%
				JSPCA (Black ink)	74.7%
Khan et al. (2)	2018	UWA Writing Ink Hyperspectral Image Database	VIS	CNN (Blue ink)	98.2%
				CNN (Black ink)	88%
Luo et al.	2015	UWA Writing Ink Hyperspectral Image Database	VIS	Own Algorithm (Blue ink)	89.0%
				Own Algorithm (Black ink)	82.3%
A. R. Martins et al.	2019	Own dataset	VNIR	UA, MCR-ALS	63%

**Table 3 sensors-22-07308-t003:** Comparison of studies (Currency).

Authors	Year	Dataset	Range (Acquisition)	Methods (Processing)	Accuracy
Baek et al.	2018	Own dataset	VNIR	PCA, SVM	99.89%
			VIS, IR	Own Algorithm	98.66%
Kang et al.	2016	Own dataset	VIS, IR	Own Algorithm	99.97%
Correia et al.	2018	Own dataset	NIR	PCA, PLS-DA	100%
Vila et al.	2006	Own dataset	IR	PCA	N/A
Lim et al.	2017	Own dataset	NIR	Own Algorithm	N/A
Hoyo-Meléndez et al.	2016	Own dataset	VNIR	Envi 5.0	N/A

**Table 4 sensors-22-07308-t004:** Comparison of studies (Photo).

Authors	Year	Dataset	Range (Acquisition)	Methods (Processing)	Accuracy
Tournié et al.	2016	Own dataset	SWIR	LDA, PCA (Agfa)	86%
				LDA, PCA (Fuji)	96.3%
				LDA, PCA (Kodak)	82.5%
Leshem et al.	2020	Own dataset	N/A	N/A	N/A
A. Martins et al.	2011	Own dataset	NIR	PCA, PLS-DA	N/A
Picollo et al.	2020	Dainelli archive	VNIR	UMAP	N/A

## Data Availability

Data sharing is not applicable.

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
