# Peer review of "Recent Advances in Counterfeit Art, Document, Photo, Hologram, and Currency Detection Using Hyperspectral Imaging"

_sensors, 2022, doi:10.3390/s22197308_

Round 1
Reviewer 1 Report
This review should address two significant issues before publication is considered. First, there are multiple grammatical errors, odd sentences, and non-statements in the text that makes it difficult to read and properly consider. Second, as a reader, I don't quite get the point of this review. It tries to summarize the studies cited while also adding a semi "statistical" comparison of the reported accuracies between studies the various studies. It really doesn't achieve either goal well. I suggest the authors concentrate on the review aspect of the manuscript. The attempt of a meta-study should be far more thorough than a comparison of reported accuracies from the methods. Many questions are left unexplored. What are the limitations of the various methods? Each of the summarized studies appears to have been examined under very limited considerations. Are these studies expected to be useful in the real world? Were there any follow up studies to each referenced method? Did other scientists build on the methods? What is the cost for each method? Are any of these systems being routinely employed to thwart counterfeits? These are just a few possible questions that could be better incorporated into the review.
Reviewer 2 Report
I find the title non-informative.
-The article is about forgery/tampering of currency and art.
-It is a literature overview
-It is about hyperspectral imaging improving upon just visual inspection.
-It shows statistics on how this hyperspectral imaging improved over the years.
The work seems to be well done but already the title and abstract are not informative to me. In the body of the paper, confusion exists on whether the choice of hyperspectral camera or or detection algorithm does the job.
I recommend a major revision keeping in mind to systematically give a clear presentation. Not confused issues.
Reviewer 3 Report
This paper developed a review related to forgery and tampering detection using HSI. The topic is interesting and meaningful. I would like to recommend to accept it after minor revision.
In Figure 1, the picture used (a mouse?) is not suitable, which is not in your topic. A document or artwork will be more beneficial.
A more comprehensive references will be better. For example, there is lots of applications in agriculture and food using HSI. In lines 59-60, you have mentioned a lot but only related to fruits and vegetables. The references of reviews are not suitable to occur in your review ([32],[36],[42], etc.). How about adding several recent research papers related to cereals (https://doi.org/10.1016/j.saa.2021.120155), meats (https://doi.org/10.1016/j.saa.2022.121689), and other aspects?
The reasons of detection accuracies in different work should be clarified. The numbers of 40%, 60%, and 99% in different publications are curious.
Reviewer 4 Report
The paper contains large and useful information, but that information is more appropriate for consumers than for designers of hyperspectral imaging systems{HIS). It seems to me that from the point of view of "Sensors" topics? there is not enough comparative analysis of the technical characteristics of HIS.
Round 2
Reviewer 2 Report
The present paper is better than version one but some small language revision is still needed.
Reviewer 4 Report
The manuscript has been sufficiently improved to publication in Sensors